# *Brucella* Shunt Infection Complicated by Peritonitis: Case Report and Review of the Literature

**Rawan Al-Qarhi** [1,2,3] **and Mona Al-Dabbagh** [1,2,4,*] 

1   Department of Paediatrics, King Abdulaziz Medical City, P.O. Box 9515, Jeddah 21423, Saudi Arabia; Rawan-Alqarhi@hotmail.com
2   King Abdullah International Medical Research Centre, P.O. Box 22384, Jeddah 22384, Saudi Arabia
3   Department of Paediatrics, Maternity and Children Hospital, Makkah 21955, Saudi Arabia
4   King Saud Bin Abdulaziz University for Health Sciences, P.O. Box 65362, Jeddah 21556, Saudi Arabia
*   Correspondence: monaaldabbagh@gmail.com; Tel.: +966-012-226-6666 (ext. 22069)

**Abstract:** Brucellosis is an endemic zoonotic disease in the Mediterranean basin and Middle East. The disease remains a diagnostic challenge due to an increasing trend of ambiguous and non-specific manifestations. We report a rare case of a 9-year-old boy who had a left frontotemporal arachnoid cyst with cystoperitoneal shunt who presented with fever for 2 weeks with gastrointestinal symptoms. He had no neurological manifestations. Diagnosis of *Brucella* shunt infection complicated with a peritoneal collection was established by isolation of the organism from cerebrospinal fluid (CSF) culture. Successful treatment was accomplished by shunt replacement and intravenous antibiotics followed by step-down oral therapy for an 18-month duration based on serological and radiological responses with no sequelae. We also reviewed the reported cases of CSF shunt infection in the literature for further guidance. *Brucella* shunt infection may be difficult to diagnose due to the diversity of signs and symptoms and the low yield on culture. Brucellosis should be always kept in mind in the differential diagnosis of patients living in endemic area with fever and non-specific symptoms. Diagnosis depends on a high index of suspicion. In addition to drug therapy, device replacement is advised to prevent treatment failure.

**Keywords:** brucellosis; neurobrucellosis and *Brucella* shunt infection

## 1. Introduction

Brucellae are Gram-negative, non-spore-forming and non-motile coccobacilli. Human brucellosis remains as one of the most common zoonotic diseases worldwide with more than 500,000 new cases annually [1]. Brucellosis is an endemic zoonotic disease in the Mediterranean basin and Middle East [2]. It is transmitted to human by contact with infected animals and consumption of raw milk and milk products. Patients typically present with recurrent fever associated with fatigue, malaise, hepatosplenomegaly, and, occasionally, spondyloarthritis. Systemic infection can rarely be complicated by specific organ involvement including infective edocarditis, nephritis, uveitis, and epidedymoorchitis. In addition, neurological involvement is considered as one of the different presentations of brucellosis assembled under the umbrella of "Neurobrucellosis" [3]. The Ministry of Health of Saudi Arabia reported an annual incidence rate of 13.41/100,000 population among Saudis in 2018 [4].

Cerebrospinal fluid (CSF) shunts are used to divert the CSF to other body parts for absorption in the setting of hydrocephalus. Infection of CSF shunts is a common complication occurring at a rate of 5–11 [5–7]. The most common aetiologic agents for shunt infections are coagulase negative Staphylococci, *Staphylococcus aureus* and *Pseudomonas aeruginosa* [8,9]. A few cases of shunt infections due to *Brucella* have been reported in the literature. Here, we intended to present a rare case of *Brucella* shunt infection and present a summary of the reported cases in the literature. We searched the PubMed database

and Google Scholar for relevant articles on "*Brucella* shunt infection" and the titles of the articles identified were reviewed. The search strategy included "*Brucella*", "Brucellosis", "neurobrucellosis", and "ventriculoperitoneal shunt" or "cerebrospinal fluid shunts" from the time the terms were introduced (starting 1963). From the literature review, we found only 12 cases of shunt infections caused by *Brucella*. The majority of the reported cases were in the paediatric age group. Our case is the third case reported from Saudi Arabia.

## 2. Case Presentation

A 9-year-old Saudi boy with a known case of left frontotemporal arachnoid cyst had a left cystoperitoneal shunt inserted 3 years prior to presentation. He presented to King Abdulaziz Medical City, Jeddah, in May 2017 with a complaint of fever and diarrhea of 2 weeks duration. The fever was subjective, undulating for 2 weeks, and subsiding transiently with antipyretic medications. The diarrhea was watery, occurred 3–4 times a day, moderate in amount, and had no blood or mucus. His symptoms progressed to abdominal pain and distention four days prior to presentation. There was no history of skin rash, headache, abnormal movements, nor loss of consciousness. No history of runny nose, cough, shortness of breath, nor cyanosis. No change in urine color, odor, nor amount. He had no history of contact with an individual diagnosed with acute gastroenteritis. No history of drug or milk allergy. He reported a history of raw camel milk ingestion many times.

Upon arrival to the emergency department, he was well looking, oriented and fairly hydrated. His temperature was 37.9 °C, heart rate 124 beats/min, blood pressure was 90/53 mmHg, and respiratory rate was 18/min. The left cystoperitoneal shunt was compressible and had no signs of malfunction. Glasgow Coma Scale score was 15/15 with no neurological deficits. He had no signs of meningeal irritation. The abdomen was mildly distended with generalized tenderness on palpation. Computed tomography (CT) of the abdomen revealed massive loculated peritoneal cavity free fluid mainly surrounding the distal tip of the VP shunt.

Blood investigations revealed the following: total leucocyte count $10.7/mm^3$, neutrophil count $8.04/mm^3$, and hemoglobin 11.8 g/dL. Renal function tests and liver function tests were normal. C-reactive protein (CRP) was 308.7 mg/L, and erythrocyte sedimentation rate (ESR) was 38 mm/h.

A CSF tap through the shunt reservoir revealed presence of white blood cells $8/mm^3$ and red blood cells $0/mm^3$. CSF protein levels were 0.322 g/L and glucose level was 1.8 mmol/L. Initially, considering the abnormal CSF findings, the patient was started on ceftriaxone and vancomycin intravenously. VP shunt removal was then undertaken, and intraabdominal cultures were obtained. CSF stain revealed presence of gram-negative coccobacilli, which grew *Brucella* species after 5 days of incubation from the CSF broth culture only. Bacterial documentation was confirmed by the Matrix-Assisted Laser Desorption Ionization (MALDI-TOF MS), which is based on protein profile identification. This was reaffirmed by the bacterial detection via an automated biotyping system to improve detection of intracellular organisms, like *Brucella* [10]. However, *Brucella* speciation was not feasible given high rate of antisera cross-reactivity.

Furthermore, *Brucella* IgM and IgG levels returned as 1.8 and 183.1 g/L, respectively. Blood and peritoneal fluid cultures was negative, and CT scan of the brain was found to be normal.

The diagnosis of *Brucella* shunt infection complicated with peritoneal collection was established by the reported history of raw milk ingestion and isolation of the organism from CSF broth culture, abnormal CSF findings, and positive serum serology for *Brucella*. Accordingly, the patient's treatment was tailored to intravenous ceftriaxone plus rifampicin and doxycycline for fourteen days accompanied with gentamicin for 1 week. The infected shunt was removed, with establishment of an external ventricular drain (EVD), followed by the placement of a new shunt when CSF sterility was achieved with antibiotic therapy.

Then, the patient was discharged on oral rifampicin and doxycycline, which were very well tolerated. He was followed up for 2 years, during which time his serum antibody titer decreased to a normal level, and the abdominal signs resolved completely based on clinical and radiological evaluations. Antibiotics were stopped after 18 months of therapy with no recurrence of the infection and a great clinical response.

## 3. Discussion

The Kingdom of Saudi Arabia (KSA) is considered as one of the highly prevalent countries with Brucellosis since the early 1980s [11,12]. Its acquisition had been linked to the traditional consumption of non-pasturized milk, mainly from camels and sheep, in addition to contact with or inhalation of infected animal products. The most commonly reported *Brucella* species in KSA include *B. melitensis*, *B. abortus* and *B. suis*, while *B. melitensis*, and *B abortus* are the most commonly reported from camels [12–14].

Brucellosis has widely variable manifestations, including fever, arthralgia, malaise and hepatosplenomegaly. It can also be present with multiorgan involvement, including arthritis, sacroiliitis, carditis, epididymo-orchitis, endophthalmitis, and neurobrucellosis [15,16]. Neurobrucellosis per se may develop at any stage of the disease. Neurobrucellosis is rare in children, and it has been reported in only 0.5–1% of children with brucellosis [17]. Most of the affected children present with acute meningitis or meningoencephalitis [17]. However, few case reports of *Brucella* VP shunt infections as a form of neurobrucellosis have been reported in the literature.

The earliest report was in 1981 by Puri et al. He reported a 5-year-old boy with a ventriculoatrial (VA) shunt who was started on empirical gentamicin and cloxacillin by the time CSF culture results came back positive for *B. abortus*. The CSF and blood became sterile, and a remarkable clinical recovery was demonstrated without specific brucellosis treatment. The author considered that the VA shunt was colonized with *B. abortus* rather than being truly infected [18]. Drutz, in 1989, presented a case of transplacentally acquired brucellosis from a mother whose pregnancy was complicated with severe joint pain, which was treated with tetracycline. Maternal dietary history revealed both prenatal and postnatal consumption of goat cheese. The child received a CSF shunt at 1 month of age and then presented at 11 months of age with CNS brucellosis. Maternal history and the early development of hydrocephalus and psychomotor delay in the infant made transplacental transmission of *Brucella* a possibility [19]. There were two cases reported from Saudi Arabia. The earlier report was in 1991 by Chowdhary et al., who concluded that treatment of patients with *B. melitensis* meningoencephalitis could be achieved with the CSF shunt system remaining in situ while systemic and intraventricular anti-*Brucella* chemotherapy is given [20]. The second case, in 2013 by Al-Otaibi et al., reported a 9-year-old child with myelomeningocele and a VP shunt who presented with gastrointestinal symptoms and peritonitis with no CNS symptoms, but the CSF was positive for *B. melitensis*. The patient did not have any neurological symptoms, the VP shunt was functioning, and CT of the brain was read as unchanged from the previous one. Therefore, the patient received intravenous doxycycline, rifampin, ciprofloxacin, and gentamicin for 2 weeks. He was discharged on an oral regimen. Two weeks later, he presented with a decreased level of consciousness, and his illness was complicated by abdominal pseudocyst formation and small bowel obstruction. Therefore, the shunt was removed and an EVD was inserted; he was placed on intravenous therapy for 6 weeks, followed by oral therapy for 10 months [21]. Al-Otaibi et al. stressed the importance of shunt replacement, in addition to drug therapy, to prevent treatment failure [21]. The rest of the reported cases are summarized chronologically in Table 1.

*Brucella* shunt infection is frequently missed due to lack of suspicion, non-specific presenting symptoms, and non-specific features on imaging. As it can be noticed from the reported cases, some patients presented with abdominal and gastrointestinal symptoms, rather than neurologic manifestations. It could be an ascending infection through the ventriculoperitoneal shunt, or vice versa, where neurobrucellosis could give rise to peritonitis.

As reported in the literature, 7 out of 12 cases of *Brucella* shunt infection had successful recovery after shunt replacement following temporary externalization, while one report did not specify if the shunt was removed or not [22]. Of the 3 cases that did not undergo shunt replacement, one case had infection relapse 2 weeks after the initial IV therapy; this was cured afterwards with shunt replacement [21]. The other two cases had successful outcomes with antibiotic therapy alone [2,20]. Two cases had infection relapse; the first one could be explained by insufficient duration of antibiotic therapy [23]. The second patient responded initially to two weeks of intravenous antibiotic therapy without removal of the shunt, but he relapsed after two weeks of oral therapy at home [21]. Only one case had psychomotor delay after the infection; this could be explained by multiple factors, such as early onset of infection through transplacental transmission, delayed recognition, diagnosis and initiation of therapy, in addition to the short treatment course for this relatively severe infection [19].

The type and duration of antimicrobial therapy was very variable; among those who had successful initial outcomes (10 cases), one was treated with shunt removal only without the use of antibiotics [18]. The overall duration of antibiotic therapy was described in the remaining 9 cases and ranged from 6 weeks to more than 6 months. The initial use of IV antibiotics was described in five cases [2,24–27], for a total duration of 2 to 6 weeks, followed by an oral regimen of 6 weeks to 6 months, while the remaining 4 cases did not report the specific method of antibiotic administration [20,22,23,28].

## 4. Conclusions

*Brucella* shunt infection may be difficult to diagnose due to the diversity of signs and symptoms and the low yield on culture. Brucellosis should always be kept in the differential diagnosis of patients coming from endemic areas with fever and non-specific symptoms or manifestations of CNS infection; diagnosis depends on a high index of suspicion. In addition to drug therapy, the device should temporarily be replaced with an EVD until the CSF is sterile and a new shunt can be placed to prevent treatment failure. Extended antimicrobial therapy beyond 6 weeks is advised; the decision on duration should be guided by clinical and microbiological responses to therapy.

**Table 1.** A summary of the previously reported cases of *Brucella* shunt infection.

| Author (Country) | Year | Age | Gender | Shunt Type | Symptoms and Signs | Lab Workup | Radiology | Treatment | Shunt Removal | Follow-up Duration | Outcome |
|---|---|---|---|---|---|---|---|---|---|---|---|
| Puri et al. (Ireland) [18] | 1981 | 5 Y | Male | VA shunt | Fever, lethargy, vomiting, headache, and skin necrosis over the valve; Hepatosplenomegaly | *B. abortus* from CSF, blood, and shunt | | Shunt removal | Yes | 2 years | Well with no sequelae |
| Drutz (Texas) [19] | 1989 | 11 M | Male | VA shunt | High *Brucella* titer, febrile infant with hydrocephalus, delayed motor skills and splenomegaly | CSF WBC: 39/mm with 12% neutrophils, protein 60 mg/dL, and glucose 51 mg/dL; *B. melitensis* grew 9 days post-op | | Tetracycline for 2 weeks, then 3 weeks of oral tetracycline | Yes | 2 years after treatment, he experienced a single prolonged tonic-clonic seizure | Psycho-motor delay |
| Chowdhary and Twum-Danso (Saudi Arabia) [20] | 1991 | 20 M | Male | VP shunt | Fever, vomiting, and difficulty in feeding. He was drowsy and could not sit or roll himself over | CSF WBC: 120 cells/mm$^3$ (polymorphonuclear cells 18% and lymphocytes 82%); low glucose and high protein; Blood culture: *B. melitensis*. *B. melitensis* agglutinin titer of 320 was positive from both CSF and serum | | Systemic and intraventricular streptomycin and rifampicin; he continued rifampicin therapy for 12 weeks after discharge | No | 12 months of follow-up | Well with no sequelae |
| Andersen et al. (Denmark) [22] | 1992 | 27 Y | Female | VP shunt | Vision disturbance, pain behind the eyes, nausea, abdominal pain and distention | Cultures from ascitic fluid, shunt system and CSF from the brain ventricles: *B. melitensis* | Abdominal US revealed a 10 × 10 cm aggregation of liquor located intraperitoneally | Ascitic fluid was drained and a fibrinous cyst was removed; tetracycline plus rifampicin for 16 weeks | Not specified | He was followed up for ten months | Well with no sequalae; abdominal and visual signs resolved completely |

**Table 1.** *Cont.*

| Author (Country) | Year | Age | Gender | Shunt Type | Symptoms and Signs | Lab Workup | Radiology | Treatment | Shunt Removal | Follow-up Duration | Outcome |
|---|---|---|---|---|---|---|---|---|---|---|---|
| Locutura et al. (Spain) [23] | 1998 | 38 Y | Male | VP shunt | Three days after placement of the shunt, he developed fever and ascites | CSF WBC: 240 cells/mm$^3$ (90%) polymorphonuclear cells) and 155 mg/dL protein; glucose level was normal; CSF and ascitic fluid cultures: *B. melitensis* | | The patient was treated for 45 days with rifampicin and doxycycline; 4 months later, infection relapse occurred in form of failure of abdominal incision healing and psychomotor agitation during the post-surgery recovery period of VP shunt replacement; treatment with rifampicin and doxycycline was prescribed again (no duration mentioned) | Yes, relapse occurred 4 months after removal | 12 months of follow-up | Good clinical evolution |
| Bessisso et al. (Qatar) [24] | 2000 | 3 Y | Female | VP shunt | Fever, weight loss, and abdominal swelling | CSF WBCs: 75/mm$^3$ (25% lymphocytes and 75% polymorphs); Low glucose and high protein; Blood and CSF cultures: *B. melitensis* | Abdominal US: cystic collection around the distal end of the VP shunt; Brain CT showed dilated ventricles and periventricular oedema | Treatment with ceftriaxone for two weeks, then rifampin and TMP-SXT for 12 weeks | Yes | Not specified | Good response to treatment |
| Alexiou et al. (Greece) [25] | 2008 | 2 Y | Male | VP shunt | High fever and signs of meningitis | CSF WBCs: 95/mm$^3$ (16% neutrophils, 62% lymphocytes); high protein and low glucose; CSF and shunt cultures: *B. melitensis* Blood culture: negative | | Treatment with rifampin and TMP-SXT for 6 weeks and gentamicin for 2 weeks | Yes | Follow-up done (no specified duration) | Good response to treatment |

**Table 1.** *Cont.*

| Author (Country) | Year | Age | Gender | Shunt Type | Symptoms and Signs | Lab Workup | Radiology | Treatment | Shunt Removal | Follow-up Duration | Outcome |
|---|---|---|---|---|---|---|---|---|---|---|---|
| Al-Otaibi et al. (Saudi Arabia) [21] | 2013 | 9 Y | Male | VP shunt | Progressive abdominal distension, vomiting, and fever | CSF WBCs: 18/mm$^3$ (88% lymphocytes and 9% polymorphs), protein 780 mg/dL, and glucose 45 mg/dL; CSF culture: *B. melitensis.* Blood and peritoneal fluid culture: negative *B. melitensis* titer in the CSF was 1:20 *B. melitensis* titer was very high at 1:20,480 | CT of the abdomen: suggestive of peritonitis with ileus; CT brain was read as unchanged from previous one; exploratory laparotomy revealed a peritoneal pseudocyst adherent to the small bowel and multiple small bowel adhesions; adhesiolysis was done | The patient received IV doxycycline, rifampin, ciprofloxacin, and gentamicin for 2 weeks. He was discharged on oral doxycycline, TMP-SXT, and rifampin; two weeks later, he had a relapse of VP shunt infection and a longer course of IV antibiotics given 6 weeks, followed by oral therapy for 10 months | Yes; Removal was after relapse of the infection | Not specified | Complete recovery |
| Abdinia et al. (Iran) [28] | 2013 | 3 Y | Male | VP shunt | Fever, ascites, vomiting, and drowsiness | CSF WBC: WBCs: 4000/mm$^3$ (70% polymorphs, 30% lymphocytes); high protein and normal glucose; CSF serological test was positive (*Brucella* agglutinin titer was 1/80) | | The patient responded well to a course of rifampin, gentamicin, and TMP-SXT daily (duration not specified) | Yes | 12 months of follow-up | Good response to treatment, with no relapse |
| Mermer et al. (Turkey) [2] | 2013 | 42 Y | Female | VP shunt | Headache, altered mental status, and convulsions | CSF WBCs: 100/mm$^3$; high protein and low glucose; 3$^{rd}$ day CSF culture: *B. melitensis* | CT brain: mild hydrocephalus | Ceftriaxone, rifampicin, and doxycycline for 6 weeks then discharged on rifampicin and doxycycline for 6 months | No | 12 months of follow-up | Good response to treatment |

**Table 1.** *Cont.*

| Author (Country) | Year | Age | Gender | Shunt Type | Symptoms and Signs | Lab Workup | Radiology | Treatment | Shunt Removal | Follow-up Duration | Outcome |
|---|---|---|---|---|---|---|---|---|---|---|---|
| Sudhamshu et al. (India) [26] | 2016 | 8 Y | Male | VP shunt | Status epilepticus with ascites and peritonitis | Blood, CSF, and ascitic cultures: *Brucella* species | | Gentamicin, rifampicin and doxycycline initially, then rifampicin and doxycycline for a total of 6 months | Yes | 12 months of follow-up | Good response to treatment |
| Mehrabian et al. (Iran) [27] | 2019 | 17 Y | Male | VP shunt | Fever, abdominal pain, and constipation. | 5th day blood culture: *Brucella* spp.; CSF culture of VP shunt: *Brucella* spp. | Abdominal US: pseudocyst at the distal end of the VP shunt; CT brain: normal | Rifampicin, TMP-SXT and ceftriaxone (IV) for 6 weeks, then discharged on rifampicin and TMP-SXT for 6 months | Yes | Not specified | Good response to treatment |

VA: ventriculoatrial; VP: ventriculoperitoneal; EVD: external ventricular drain; CSF: cerebrospinal fluid; IV: intravenous; TMP-SXT: trimethoprim-sulfamethoxazole.

**Author Contributions:** R.A.-Q. wrote the introduction, case report, literature summary, summary table and discussion; M.A.-D. had the idea, conducted the literature search, supervised the project, edited the manuscript, and wrote part of the discussion. All authors have read and agreed to the published version of the manuscript.

**Funding:** This research received no external funding.

**Institutional Review Board Statement:** Reporting of this case was approved by the institutional review board (IRB) of King Abdullah International Medical Research Centre prior to submission, with IRB approval number: RJ20/034/J. As per IRB recommendations, consent for participation was not required; however, consent for publication was obtained from the parents upon journal submission.

**Informed Consent Statement:** A written informed consent was obtained from the patient's father to publish this paper.

**Data Availability Statement:** Not applicable.

**Acknowledgments:** Special thanks to the paediatric infectious disease team that contributed to the management of this patient during follow-up visits. The authors would also like to thank Adulfattah Al Amri for his feedback re-microbiological diagnosis.

**Conflicts of Interest:** The authors declare no conflict of interest.

## Abbreviations

| | |
|---|---|
| CSF | Cerebrospinal fluid |
| EVD | External ventricular drainage |
| CT | Computed tomography |
| CRP | C-reactive protein |
| ESR | Erythrocyte sedimentation rate |
| VP | Ventriculoperitoneal |
| VA | Ventriculoatrial |
| CNS | Central nervous system |
| KSA | Kingdom of Saudi Arabia |

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
