# Peer review of "Brucella Shunt Infection Complicated by Peritonitis: Case Report and Review of the Literature"

_2036-7449, doi:10.3390/idr13020035_

Round 1
Reviewer 1 Report
Even though similar cases were reported, this is an interesting case report about neurobrucellosis diagnosed an finally proven by CSF culture.
Author Response
The authors would like to thank the reviewer for the valuable comments and constructive feedback.
English language was revised
Additional edits:
First author’s department was added
Acknowledgement section was updated
Abbreviations were updated
All changes in the manuscript are written in red

Reviewer 2 Report
"Brucella shunt infection complicated by peritonitis: case report and review of the literature" is an interesting manuscript that describes a case of shunt infection by Brucella spp. and then reviews 12 other known cases that are similar. The authors do a good job in describing the case and briefly reviewing the common difficulties in the diagnosis of brucellosis.
Major Comment:
There is no discussion of the Brucella spp. that was lab cultured in this case. If the species was undetermined for some reason that should be discussed and then there should be a discussion of the most likely spp. For instance, was the patient consuming cow or goat's milk regularly?
The authors claim that brucellosis is the "most common zoonotic infection" is not correct. Please re-word to "brucellosis is one of the most common zoonotic infections." Page 1 Line 32
Minor Comments:
There are a few grammatical errors in the manuscript. Please review for grammar before publication. For example Page 1 Line 17: "He had no any neurological manifestations." would be better as " He had no neurological manifestations."
When describing the "subjective, on and off" nature of the fever in the patient, please include the term "undulating fever" for clarity and as a reference to a known common symptom of brucellosis.
Author Response
Major Comment:
- There is no discussion of the Brucella spp. that was lab cultured in this case. If the species was undetermined for some reason that should be discussed and then there should be a discussion of the most likely spp. For instance, was the patient consuming cow or goat's milk regularly?
Response
A detailed paragraph was added on bacterial detection techniques in line 87. A word ”camel” was added in line 68 to specify the source of the consumed milk.
Furthermore, a paragraph was added in the discussion section on brucella epidemiology in KSA including details on the common Brucella species acquired from Camels.
- The authors claim that brucellosis is the "most common zoonotic infection" is not correct. Please re-word to "brucellosis is one of the most common zoonotic infections." Page 1 Line 32
Response: Modification made accordingly
Minor Comments:
- There are a few grammatical errors in the manuscript. Please review for grammar before publication. For example, Page 1 Line 17: "He had no any neurological manifestations." would be better as " He had no neurological manifestations."
Response: Grammar check was done
- When describing the "subjective, on and off" nature of the fever in the patient, please include the term "undulating fever" for clarity and as a reference to a known common symptom of brucellosis.
Response: Text modified as suggested
- Additional edits:
First author’s department was added
Acknowledgement section was updated
Abbreviations were updated
All changes in the manuscript are written in red

Reviewer 3 Report
General considerations:
The authors report a case of Brucella infection from a CSF shunt and say they review the cases from the CSF shunt literature. On the one hand, they do not indicate how they typified Brucella (lines 78-80), then it is not reliably appreciated that their isolate is indeed Brucella. And on the other hand, when they indicate that they did the bibliography search, they do not indicate the years that it covered (lines 44-48). The authors should mention both data.
In my opinion, the authors make a very brief review of the bibliography. When I have done a search in my database for "Neurobrucellosis", I have found all the citations that I provide in the attached document.
As can be seen, neurobrucellosis occurs from time to time in humans, especially in children and it has also been described in marine mammals, so the isolation of Brucella from CSF and contamination with this pathogen of its drains and cochlear implants should be taken into account in regions where the disease has a high incidence. I believe that the authors should direct more efforts to this fact.
Specific considerations
Some references in Table 1 are incorrect:
Where it says (9) should it say (12)
Where it says (10) should it say (13)
Where it says (11) should it say (16)
Where it says (12) should it say (15)
Where it says (13) should it say (17)
Where it says (14) should it say (18)
Where it says (15) should it say (19)
Where it says (16) should it say (14)
Where it says (17) should it say (22)
Where it says (18) should it say (20)
Where it says (19) should it say (21)
Reference 14 should be name at line 120 after Al-Otaibi et al.
Reference 15 should be name before than 16 in line 159

Author Response
The authors would like to thank the reviewers for the valuable comments and constructive feedback. Below are the authors responses:
- The authors report a case of Brucellainfection from a CSF shunt and say they review the cases from the CSF shunt literature. On the one hand, they do not indicate how they typified Brucella (lines 78-80), then it is not reliably appreciated that their isolate is indeed Brucella.
Response: A detailed paragraph was added on bacterial detection techniques in line 87.
- And on the other hand, when they indicate that they did the bibliography search, they do not indicate the years that it covered (lines 44-48). The authors should mention both data.
Response: The years covered were added accordingly
- In my opinion, the authors make a very brief reviewof the bibliography. When I have done a search in my database for "Neurobrucellosis", I have found all the citations that I provide in the attached document.
As can be seen, neurobrucellosis occurs from time to time in humans, especially in children and it has also been described in marine mammals, so the isolation of Brucella from CSF and contamination with this pathogen of its drains and cochlear implants should be taken into account in regions where the disease has a high incidence. I believe that the authors should direct more efforts to this fact.
Response: Thank you for your feedback, however, the focus of the paper was specific to brucella associated VP shunt infection and not all neurobrucellosis or those associated with cohlear implants. This in fact was the unique thing about the paper as Brucella shunt infection was not reported in many studies. The keyword "Neurobrucellosis" was added to our search strategy because some of the articles reporting shunt infection used it, thus we wanted to make sure that the search strategy used was indeed exhaustive. In addition, the fact that Brucella should be added to the differential diagnosis in endemic areas was actually our main conclusion that we stressed on in both the abstract and conclusion. A short sentence was added in the conclusion to include manifestations of CNS infection as one of the symptoms deserving high index of suspension.
Specific considerations
Some references in Table 1 are incorrect:
Where it says (9) should it say (12)
Where it says (10) should it say (13)
Where it says (11) should it say (16)
Where it says (12) should it say (15)
Where it says (13) should it say (17)
Where it says (14) should it say (18)
Where it says (15) should it say (19)
Where it says (16) should it say (14)
Where it says (17) should it say (22)
Where it says (18) should it say (20)
Where it says (19) should it say (21)
Reference 14 should be name at line 120 after Al-Otaibi et al.
Reference 15 should be name before than 16 in line 159
Response: Thank you for your observation and great feedback. It seems that the references in the table changed upon inserting the table in the text template provided by the journal. All references were corrected as advised.
Additional edits:
First author’s department was added
Acknowledgement section was updated
Abbreviations were updated
All changes in the manuscript are written in red
Reviewer 4 Report
The work is very interesting, due to the rarity of shunt infection involvement for Brucella spp and the relevance of that pathogen in many areas worldwide. Some minor revision could add value to the work, in my opinion; some suggestions:
- at line 38, a brief overview for Brucella infection and clinical presentation, before introducing cerebro-spinal shunts
- at line 78, a brief exposure about differential diagnosis process
- table 1: the text could be summarized (e.g. for Lab workup: avoid verbs to reduce the text) to better the effectiveness of presenting data
- a light revision of text editing
Author Response
-
- At line 38, a brief overview for Brucella infection and clinical presentation, before introducing cerebro-spinal shunts
Response: Briefing of the clinical presentation of Brucella was added in the corresponding section
- At line 78, a brief exposure about differential diagnosis process
Response: more details were added on the diagnostic workup done for the patient
- Table 1: the text could be summarized (e.g. for Lab workup: avoid verbs to reduce the text) to better the effectiveness of presenting data
Response: Many of the verbs in the lab workup column were removed and the sentences were shortened.
- A light revision of text editing
Response: Modified as appropriate
-Additional edits:
First author’s department was added
Acknowledgement section was updated
Abbreviations were updated
All changes in the manuscript are written in red

Round 2
Reviewer 3 Report
The authors have sufficiently satisfied the indications made previously.
Three small details remain to be corrected:
In Table 1 where it say "Sudhamshu" should it say "Sudhamshu et al."
In line 268 where it say "PURI P" should it say "P. Puri"
In line 273 where it say "A. AL-otaibi" should it say "A. AL-Otaibi"
Author Response
The authors would like to thank the reviewers for the valuable comments and constructive feedback. Below are the authors responses:
Three small details remain to be corrected:
In Table 1 where it say "Sudhamshu" should it say "Sudhamshu et al."
In line 268 where it say "PURI P" should it say "P. Puri"
In line 273 where it say "A. AL-otaibi" should it say "A. AL-Otaibi"
Response:
Thank you for your observation and great feedback. All suggestions were modified as advised.
All changes are written in red font
